# Clinical Activity of an hTERT-Specific Cancer Vaccine (Vx-001) in “Immune Desert” NSCLC

**DOI:** 10.3390/cancers13071658

**Published:** 2021-04-01

**Authors:** Ioannis S. Pateras, Athanasios Kotsakis, Margaritis Avgeris, Evangelia Baliou, Panagiotis Kouroupakis, Eleni Patsea, Vassilis Georgoulias, Jeanne Menez-Jamet, Jean-Pierre Kinet, Kostas Kosmatopoulos

**Affiliations:** 1Department of Histology and Embryology, Medical School, National and Kapodistrian University of Athens, 11527 Athens, Greece; ipateras@med.uoa.gr; 2Department of Medical Oncology, University General Hospital of Larissa, 41110 Larissa, Greece; thankotsakis@uth.gr; 3Laboratory of Clinical Biochemistry—Molecular Diagnostics, 2nd Department of Pediatrics, School of Medicine, National and Kapodistrian University of Athens, 11527 Athens, Greece; margaritis.avgeris@gmail.com; 4Department of Biochemistry and Molecular Biology, Faculty of Biology, National and Kapodistrian University of Athens, 15771 Athens, Greece; 5Department of Pathology, Athens Medical Center, 15126 Marousi, Greece; vbaliou@yahoo.com; 6Department of Hematology, Sismanogleion General Hospital of Athens, 15126 Marousi, Greece; kouroupp@gmail.com; 7Department of Pathology, Metropolitan Hospital, 18547 Cholargos, Greece; elenipats@gmail.com; 8Hellenic Oncology Research Group, 11474 Athens, Greece; georgulv@otenet.gr; 9Vaxon Biotech, 75005 Paris, France; jmenez@vaxon-biotech.com (J.M.-J.); jean-pierre.kinet@ixlife.com (J.-P.K.); 10Department of Pathology, Harvard Medical School, Boston, MA 02215, USA

**Keywords:** cancer vaccines, Vx-001, metastatic non-small cell lung cancer, immunologically cold tumors, tumor-infiltrating lymphocytes, granzyme B

## Abstract

**Simple Summary:**

We investigated whether there is any correlation between Vx-001 clinical activity and the tumor immune microenvironment (TIME). Our hypothesis was that Vx-001 should be clinically effective in patients with tumor-infiltrating lymphocyte (TIL) negative/low infiltrated (non-immunogenic/cold) tumors which are lacking immunosuppressive TIME but not in highly TIL infiltrated (immunogenic/hot) tumors associated with immunosuppressive TIME. In this study, we show that the tumor vaccine Vx-001 offers a clinical benefit in patients with tumors lacking or weakly infiltrated with TILs. In contrast, Vx-001 is completely inactive in the context of tumors highly infiltrated with TILs, thus confirming our hypothesis. TIL negative/low tumor signature is an independent predictive factor of Vx-001 efficacy. To our knowledge, this is the first study showing an inverse correlation between tumor vaccine efficacy and the presence of TILs. These data support the selection of patients with TIL negative or low infiltrated tumors (i.e., patients known to be resistant to immune checkpoint inhibitors (ICIs) and with poor prognosis) as the best candidates to receive tumor vaccines and to get a clinical benefit from vaccination.

**Abstract:**

Background: Tumors can be separated into immunogenic/hot and non-immunogenic/cold on the basis of the presence of tumor-infiltrating lymphocytes (TILs), the expression of PD-L1 and the tumor mutation burden (TMB). In immunogenic tumors, TILs become unable to control tumor growth because their activity is suppressed by different inhibitory pathways, including PD-1/PD-L1. We hypothesized that tumor vaccines may not be active in the immunosuppressive microenvironment of immunogenic/hot tumors while they could be efficient in the immune naïve microenvironment of non-immunogenic/cold tumors. Methods: The randomized phase II Vx-001-201 study investigated the effect of the Vx-001 vaccine as maintenance treatment in metastatic non-small cell lung cancer (NSCLC) patients. Biopsies from 131 (68 placebo and 63 Vx-001) patients were retrospectively analyzed for PD-L1 expression and TIL infiltration. TILs were measured as tumor-associated immune cells (TAICs), CD3-TILs, CD8-TILs and granzyme B-producing TILs (GZMB-TILs). Patients were distinguished into PD-L1(+) and PD-L1(-) and into TIL high and TIL low. Findings: There was no correlation between PD-L1 expression and Vx-001 clinical activity. In contrast, Vx-001 showed a significant improvement of overall survival (OS) vs. placebo in TAIC low (21 vs. 8.1 months, *p* = 0.003, HR = 0.404, 95% CI 0.219–0.745), CD3-TIL low (21.6 vs. 6.6 months, *p* < 0.001, HR = 0.279, 95% CI 0.131–0.595), CD8-TIL low (21 vs. 6.6 months, *p* < 0.001; HR = 0.240, 95% CI 0.11–0.522) and GZMB-TIL low (20.7 vs. 11.1 months, *p* = 0.011, HR = 0.490, 95% CI 0.278–0.863). Vx-001 did not offer any clinical benefit in patients with TAIC high, CD3-TIL high, CD8-TIL high or GZMB-TIL high tumors. CD3-TIL, CD8-TIL and GZMB-TIL were independent predictive factors of Vx-001 efficacy. Conclusions: These results support the hypothesis that Vx-001 may be efficient in patients with non-immunogenic/cold but not with immunogenic/hot tumors.

## 1. Introduction

Until now, cancer vaccines have largely failed because they targeted tumor self-antigens and were not able to overcome immune tolerance [1]. Overcoming tolerance to tumor antigens without inducing autoimmunity is therefore one of the main challenges in tumor vaccination.

Cryptic peptides from tumor self-antigens are blind spots of the immune system. They have low affinity for the major histocompatibility complex (MHC), and because they are not presented efficiently by the MHC, they are not seen by the immune system. Thus, they lack both the capacity to generate an immune response and to induce self-tolerance [2,3]. Cryptic peptides can be engineered to have high affinity for antigen presentation molecules (optimized cryptic peptides). Consequently, these optimized cryptic peptides become immunogenic while still bypassing the immune tolerance network. The optimized cryptic peptide strategy is therefore a potential novel approach in cancer vaccination. Preclinical and clinical studies with the optimized cryptic peptide vaccine Vx-001, that targets the universal tumor antigen telomerase reverse transcriptase (TERT) have revealed its capacity to induce a cross-reactive immune response against the wild type (WT) cryptic peptide, which in turn can be amplified further with the WT peptide. This amplification with WT cryptic peptide is possible in spite of its low affinity for the MHC because the cross-reactive CD8 lymphocytes help maintain the WT peptide in the MHC groove, thereby permitting the amplification of the response [3,4,5].

In a phase I/II study, Vx-001 induced objective responses and long-lasting disease control (>12 months) in patients with metastatic tumors who were progressing before vaccination [6,7,8]. Long-lasting responses were significantly associated with the absence of pre-vaccine endogenous TERT-specific tumor immunity, suggesting that long-lasting responses may only be observed in patients with low- or non-immunogenic tumors (unpublished data). This could probably explain why Vx-001 failed to improve overall survival (OS) of unselected patients when tested as maintenance treatment in the randomized Vx-001-201 phase II study in metastatic non-small cell lung cancer (NSCLC) patients who experienced disease control after first line chemotherapy [9]. Sub-group analysis of this study suggested that Vx-001 is active in never/light smokers and in patients without pre-vaccination endogenous antitumor immunity, two criteria which, taken together, could be considered to be associated with the presence of non-immunogenic tumors [10].

During the last few years, it became clear that there is a heterogeneity of the tumor immune landscape. Indeed, it has been shown that tumors can be divided into immunogenic/hot, which are sensitive to immune checkpoint inhibitors (ICIs), and non-immunogenic/cold, which are considered to be resistant to ICIs. Immunogenic/hot tumors which, in general, harbor a high frequency of neoantigens, can be defined by the presence of tumor-infiltrating lymphocytes (TILs), the expression of programmed death ligand 1 (PD-L1) in tumor cells or/and immune cells and a high tumor mutation burden (TMB) [11].

Patients with TIL(+) immunogenic/hot tumors develop a polyspecific endogenous neoantigen-specific immune response that becomes unable to control tumor growth mainly because of the different inhibitory mechanisms, including the PD-1/PD-L1 pathway.

Based on these assumptions, it could be hypothesized that the mono- or oligo-specific immune responses, generated by a specific vaccine in the immunosuppressive microenvironment of TIL(+) tumors, is unlikely to offer a clear clinical benefit. In contrast, vaccines may be effective in the absence of the immunosuppressive microenvironment seen in immunologically cold TIL(-) tumors. These tumors are non-immunogenic, harboring low TMB and few neoantigens, and they are resistant to ICIs [12,13].

In the present study, we assessed TIL infiltration and PD-L1 expression of primary tumors in the NSCLC patients enrolled in the Vx-001-201 study in order to investigate whether patients with low-TIL or PD-L1(-) tumors might be more responsive to the Vx-001 vaccine. Our results showed that vaccination with Vx-001 was associated with a significantly better overall survival (OS) and time to treatment failure (TTF) compared to placebo in patients with low-TIL tumors but not in patients with high-TIL tumors.

## 2. Materials and Methods

### 2.1. Patients and Study Design

A detailed design of the multicenter, double blind, placebo controlled, randomized phase IIb Vx-001-201 clinical study has been recently described (12; clinicaltrials.gov: NCT01935154)). The aim of the study was to evaluate the role of the Vx-001 cancer vaccine as maintenance immunotherapy in metastatic NSCLC patients who experienced disease control after 1st line chemotherapy. The main eligibility criteria were histologically documented metastatic NSCLC, performance status (Eastern Cooperative Oncology Group; ECOG) of 0–1, HLA-A*0201 haplotype and tumoral expression of TERT. Patients were randomly assigned (1:1) to receive six vaccinations of Vx-001 or placebo every 3 weeks and then one vaccination every three months. Patients discontinued vaccination because of disease progression according to investigator evaluation, toxicity, consent withdrawal or death. The primary objective of the study was the OS and the secondary objective was the time to treatment failure (TTF) in Vx-001-treated vs. placebo-treated patients

### 2.2. PD-L1 Staining

PD-L1 expression was assessed at Sciempath Bio (France) with the PD-L1 IHC 22C3 pharmDx assay (Agilent Technologies, Carpinteria, CA, USA) and measured in formalin-fixed tumor samples obtained by core-needle or excisional biopsy of the primary tumor lesion at diagnosis. Expression was categorized by the tumor proportional score (TPS), which was defined as the percentage of tumor cells with membranous PD-L1 staining [14]. Patients were classified in two groups (i) PD-L1 negative (PD-L1(-); TPS < 1%) and (ii) PD-L1 positive (PD-L1(+); TPS > 1%).

### 2.3. Lymphocytic Infiltration

Tumor-associated immune cells (TAICs) were qualitatively evaluated on hematoxylin and eosin (H&E)-stained sections. The TAIC infiltration was evaluated for both intra-tumoral and stromal localization. The grading system was absent (0)/minimal (1+)/mild (2+)/moderate (3+)/marked (4+) and was based on the extent and severity of the infiltration [15]. Tumors with score 0 and 1+ were considered as low TAIC while tumors with a score of 2+, 3+ and 4+ were characterized as high TAIC. Furthermore, TILs were evaluated by measuring CD3 positive, CD8 positive and granzyme B (GZMB) positive cells. The phenotypic characterization of TILs was immunohistochemically (IHC) assessed by incubating Formalin Fixed Paraffin Embedded (FFPE) tumor sections with the following antibodies: anti-CD3 (2GV6, Ventana, Roche, Oro Valley, AZ, USA), anti-CD8 (C8/144B, Cell Marque, Rocklin, CA, USA) and anti-granzyme B (ab4059, Abcam, UK). For IHC analysis, the Ventana Benchmark XT stainer was used. For CD3, CD8 and GZMB evaluation, the number of positive cells within a tumor per high-power field (HPF; magnification, 400×) in the entire tissue section was counted and the mean value was extracted, as previously described [16]. A normal human lymph node served as a poitive control for CD3, CD8 and GZMB. For double CD8 and GZMB immunostaining, CD8 was visualized in brown and GZMB in red. The evaluation of double immunostaining was performed based on the ratio (CD8+GZMB+)/(CD8+) per HPF. Slide assessment was performed by three independent observers (EB, EP and ISP) with minimal inter-observer variability.

### 2.4. Immune Response

A vaccine-induced immune response was evaluated before the first vaccination (baseline), before the third vaccination (W6) and at week 18 or at the end of treatment visit for patients who dropped out before the sixth vaccination. Thereafter, from week 39, it was evaluated every 24 weeks.

An IFNg ELISpot assay was performed using the Human IFN-g ELISpot PVDF-Enzymatic kit (Diaclone, Besançon, France) as described previously [9].

### 2.5. Statistical Analysis

Statistical analysis was performed by IBM SPSS Statistics 20 software (IBM Corp., Armonk, NY, USA). The normal distribution of the data was tested by Shapiro–Wilk and Kolmogorov–Smirnov tests. The association of categorical data was assessed by a Pearson chi-square test, while the correlation of continuous variables was evaluated by Spearman correlation. The X-tile algorithm [17] was applied for the selection of the optimal cut-off values for CD3-TIL, CD8-TIL and GZMB-TIL levels, equal to 18, 9 and 3 positive cells/HPF, respectively. These cut-off values distinguish low-TIL from high-TIL tumors. Survival analysis was performed by Kaplan–Meier curves using a log-rank test, and Cox proportional regression analysis. Bootstrap Cox proportional regression models based on 1000 bootstrap samples were used for internal validation. Patient’s death and treatment failure were used as clinical endpoint events.

## 3. Results

### 3.1. TIL Phenotype and PD-L1 Expression

One hundred and thirty-one (68 placebo and 63 Vx-001) patients with available tissue from the primary tumor biopsy were included in the study and Table 1 shows their clinicopathologic characteristics.

All patients were tested for TAIC and PD-L1 expression. Seventy-one (54.2%) of them were characterized as TAIC-high and 30 (22.9%) as PD-L1(+). Furthermore, TAICs were phenotypically characterized as CD3-TIL (*n* = 105 patients) and CD8-TIL (*n =* 101 patients), whereas granzyme B (GZMB) expression, which is one of the major cytotoxic molecules released by cytotoxic TILs, was assessed in 85 patients. Sixty (57.1%) patients were characterized as CD3-TIL high, 59 (58.4%) as CD8-TIL high and 15 (17.6%) as GZMB-TIL high (Figure 1).

Moreover, double CD8/GZMB staining, which was performed in 30 patients, revealed that 27 (90%) of them did not harbor CD8(+)/GZMB(+) TILs (Appendix A). TAIC high tumors were significantly associated with an increased percentage of CD3-TIL high (*p* < 0.001) and CD8-TIL high (*p* < 0.001) compared to TAIC low tumors (Appendix A). Spearman analysis highlighted the significant correlation of CD3-TIL and CD8-TIL (r_s_: 0.868; *p* < 0.001; Appendix A). A significant correlation of GZMB-TIL and CD3-TIL (r_s_ = 0.392; *p* < 0.001) or CD8-TIL (r_s_ = 0.416; *p* < 0.001) was also observed (Appendix A). No significant correlation was observed between PD-L1, TAIC, CD3-TIL, CD8-TIL and GZMB-TIL and gender, age, response to previous treatment, histology and smoking status (Appendix A). The groups of patients enrolled in the two treatment arms were well balanced (Appendix A).

### 3.2. TILs but Not PD-L1 Predict Vx-001 Efficacy on OS

The effect of the Vx-001 vaccine on OS was evaluated according to the expression of PD-L1 on tumor cells. The survival analysis showed that PD-L1 is not a discriminatory factor since Vx-001 could not significantly improve OS in patients with either PD-L1(-) (9.9 vs. 16.8 months, *p* = 0.09; (Figure 2A)) or PD-L1 (+) tumors (Appendix A) compared to placebo. Conversely, OS was significantly prolonged in Vx-001-treated compared to placebo-treated patients within TAIC low (8.1 vs. 21 months, *p* = 0.003; HR = 0.404, 95% CI 0.219–0.745; (Figure 2B)), CD3-TIL low (6.6 vs. 21.6 months, *p* < 0.001; HR = 0.279 95% CI 0.131–0.595; (Figure 2C)) and CD8-TIL low (6.6 vs. 21 months, *p* < 0.001; HR = 0.240, 95% CI 0.111–0.522; (Figure 2D)) sub-cohorts.

In contrast, Vx-001 did not prolong OS for the patients with TAIC high, CD3-TIL high and CD8-TIL high tumors (Appendix A).

### 3.3. TILs but Not PD-L1 Predict Vx-001 Efficacy on TTF

Vx-001 did not demonstrate any effect on TTF in PD-L1(-) patients (2.3 vs. 3.4 months, *p* = 0.08) (Figure 3A). Further analysis of Vx-001’s effect on TTF demonstrated that Vx-001 significantly prolonged TTF compared to placebo in patients with a TAIC low (2.7 vs. 3.6 months, *p* = 0.042; HR = 0.583, 95% CI 0.336–1.013; (Figure 3B)), CD3-TIL low (2.1 vs. 3.6 months, *p* = 0.002; HR = 0.360, 95% CI 0.181–0.719; (Figure 3C)) or CD8-TIL low (2 vs. 3.9 months, *p* = 0.001; HR = 0.333, 95% CI 0.158–0.700; (Figure 2D)) phenotype.

In contrast, Vx-001 did not show any effect on TTF in patients harboring PD-L1(+), TAIC high, CD3-TIL high or CD8-TIL high tumors (Appendix A).

### 3.4. Functionality of TILs Predicts Activity of Vx-001 on Both OS and TTF

Seventy patients with GZMB-TIL low and 15 patients with GZMB-TIL high tumors were further analyzed in order to investigate any relation between the functionality of TILs and activity of Vx-001. Thirty nine out of 68 (57.4%) patients with GZMB-TIL low tumors were classified as CD8-TIL high. Vx-001 significantly improved OS (11.1 vs. 20.7 months, *p* = 0.011; HR = 0.490, 95% CI 0.278–0.863; (Figure 4A)) and TTF (2.3 vs. 3.5 months, *p* = 0.023, HR = 0.575, 95% CI 0.346–0.955; (Figure 4B)) in patients with GZMB-TIL low but not in patients with GZMB-TIL high tumors, although the number of GZMB-TIL high patients was small (*n =* 15) (Appendix A).

This observation was further supported by the findings in the sub-group of patients with tumors harboring double negative CD8/GZMB TILs (OS: 9.3 vs. 21 months *p* = 0.003, HR = 0.351, 95% CI 0.171–0.721 and TTF: 2 vs. 3.6 months, *p* = 0.002, HR = 0.356, 95% CI 0.176–0.722) (Appendix A).

### 3.5. Vaccine Induced Immune Response and TIL Infiltration

The TERT_572_-specific immune response was tested in 53 placebo-treated and 52 Vx-001-treated patients. Seventeen (32.7%) patients in the Vx-001 arm developed an immune response. An immune response was not detected in the placebo arm. There was no correlation between detected immune response and the CD3/CD8-TIL profile of tumors. Additionally, 40.9% (9/22) of patients with CD3/CD8-TIL low tumors and 25% (6/24) of patients with CD3/CD8-TIL high tumors responded to Vx-001 (*p* = 0.348). Furthermore, the correlation of immune response and OS was investigated in patients with CD3/CD8-TIL low and CD3/CD8-TIL high tumors. Although the number of patients is low, the results suggested that immune response was correlated with longer OS in patients with CD3/CD8-TIL low tumors but not in patients with CD3/CD8-TIL high tumors (Appendix A).

#### Univariate and Multivariate Analysis

Univariate Cox regression analysis confirmed the improved OS of the CD3-TIL low (HR = 0.395; 95% CI: 0.194–0.805; *p* = 0.011), CD8-TIL low (HR = 0.428; 95% CI: 0.212–0.862; *p* = 0.018) and GZMB-TIL low (HR = 0.310; 95% CI: 0.122–0.792; *p* = 0.014) patients treated with Vx-001. Conversely, a similar association could not be observed in TAlC low or TAIC high patients as well as in the other patient subgroups (Table 2). Additionally, using multivariate Cox models adjusted for patients’ gender, age, smoking status, tumor histology and response to previous treatment, CD3-TIL low (HR = 0.381; 95% CI: 0.181–0.799; *p* = 0.011), CD8-TIL low (HR = 0.388; 95% CI: 0.182–0.830; *p* = 0.015) and GZMB-TIL low (HR = 0.173; 95% CI: 0.058–0.520; *p* = 0.002) emerged as independent predictive factors associated with improved OS in patients treated with Vx-001 (Table 2).

## 4. Discussion

In the current study, we investigated whether there is a correlation of clinical efficacy of Vx-001 with the tumor immune microenvironment (TIME) as defined by the PD-L1 expression and the TIL infiltration. Our hypothesis was that the mono- or oligo-specific immune response to the hTERT antigen, generated by Vx-001, was unlikely to offer a significant clinical benefit in the context of the immunosuppressive microenvironment of TIL(+) immunogenic tumors. In fact, patients with TIL(+) immunogenic tumors have already been found to develop a polyspecific endogenous neoantigen-specific immune response that is unable to control tumor growth, mainly because of the different inhibitory mechanisms, including the PD-1/PD-L1 pathway. In contrast, Vx-001 may offer a clinical benefit to patients with low-TIL tumors which are known as an “immune desert” and are lacking such an immunosuppressive microenvironment [18,19]. Our results support this hypothesis and show that vaccination with Vx-001 was associated with significant clinical benefit, in terms of both TTF and OS, in patients with TIL low but not TIL high tumors.

In our study, TIL infiltration was evaluated by measuring TAIC, CD3-TIL and CD8-TIL. These three approaches gave comparable results. There was a very strong correlation between TAIC high, CD3-TIL high and CD8-TIL high tumors. Vx-001 significantly improved both OS and TTF in patients with TIL(-) tumors but not in patients with TIL(+) tumors. Importantly, CD3-TIL and CD8-TIL infiltration emerged as independent predictive factors of Vx-001’s clinical benefit in multivariate analysis adjusted for patients’ gender, age, smoking status, tumor histology and response to previous treatment. TILs are mostly specific to tumor neoantigens and their presence is associated with high TMB, leading to immunogenic neoantigens, thus characterizing the immunogenic tumors [20,21]. Several reports highlight the role of TILs as prognostic factors in NSCLC and concluded that TIL infiltration is an independent prognostic factor for NSCLC [15,21,22,23]. Moreover, several studies reported that TIL infiltration is associated with responsiveness to ICIs [13,23,24]. The predictive value of TIL infiltration regarding the clinical efficacy of ICIs is easily explained by the mode of action of ICIs. The role of ICIs is to release the activity of TILs by blocking the inhibitory pathways, including PD-L1/PD-1, developed by the tumor in order to escape the endogenous tumor immunity. The presence of TILs seems therefore to be a prerequisite for ICI efficacy.

Most tumors in our study were GZMB-TIL low. Moreover, CD8-TILs were GZMB(-) in 77.8% of patients and double positive CD8/GZMB TILs were not detected in 90% (27/30) of patients. The phenotype of GZMB(+)/CD8(-) TILs was not studied but it is very likely that they are Natural Killer (NK) cells. These findings are in line with other reports showing that CD8-TILs in NSCLC do not produce GZMB, probably because of activation-induced cell death (AICD) or because of soluble factors secreted by the tumor, such as PGE2 [25,26]. It does not seem that the CD8 inactivation, as assessed by the absence of GZMB production, is due to the PD-1/PD-L1 inhibitory pathway because we did not find any correlation between GZMB-TIL frequency and the tumoral expression of PD-L1. In the multivariate analysis, GZMB-TIL low emerged as an independent predictive factor associated with better OS in Vx-001-treated patients. The clinical benefit of Vx-001 in patients with GZMB-TIL low tumors cannot be explained by the low frequency of CD8-TIL in the GZMB-TIL low tumors. In fact, more than 50% of GZMB-TIL low tumors were CD8-TIL high. It would be interesting to study additional markers of T cell activation such as IFNg and perforin to investigate if the activity of Vx-001 is related not only to the number of TILs but also to their functionality. Little is known about the role of GZMB in the responsiveness to ICIs. Elevated levels of serum GZMB have been shown to be associated with a better response to ICIs compared to low levels of serum GZMB [27,28].

The results of our study could explain the poor efficacy of Vx-001 in the unselected NSCLC population observed in the randomized phase II Vx-001-201 study since less than 50% of the enrolled patients had TIL low tumors [12]. They could also explain why the absence of PD-L1 expression is not associated with Vx-001 efficacy since 48.1% and 53.9% of PD-L1(-) tumors were CD3-TIL high and CD8-TIL high, respectively, in agreement with the literature data showing that 40% of PD-L1(-) NSCLCs are classified as TIL(+) [18].

It is tempting to speculate that our findings with Vx-001 could be extended to all cancer vaccines that are able to induce a strong anti-tumor immunity. Thus, cancer vaccines should be tested in patients selected for their low-TIL, non-immunogenic NSCLC tumors that are resistant to ICIs and not in patients with high-TIL, immunogenic NSCLC known to be sensitive to ICIs.

Whether the above conclusions in the context of NSCLC could be extrapolated to other “immune desert” tumors such as microsatellite stable (MSS) tumors remains to be investigated in future studies. In fact, more than 70% of MSS colorectal and MSS gastric cancers are low TIL [29]. It is well known that these MSS tumors are resistant to ICIs and have poor prognosis [29].

Although the Vx-001-201 study was not initially designed to evaluate the efficacy of Vx-001 in non-immunogenic/cold NSCLC, our results strongly suggest that Vx-001 is clinically active only in this specific patient population. A prospective study is being planned to confirm the above findings and to further address the question of whether Vx-001 can turn non-immunogenic cold NSCLCs that are resistant to ICIs into hot NSCLCs that are sensitive to ICIs.

## 5. Conclusions

In this study, we show that the tumor vaccine Vx-001 offers a clinical benefit to patients with tumors lacking or weakly infiltrated with TILs but not to patients with tumors highly infiltrated with TILs. The TIL negative/low tumor signature is an independent predictive factor of Vx-001 efficacy. To our knowledge, this is the first study showing an inverse correlation between tumor vaccine efficacy and the presence of TILs. These data support the selection of patients with TIL negative or poorly infiltrated tumors (i.e., patients known to be resistant to ICIs and with bad prognosis) to be the best candidates to show a clinical benefit from vaccination.

## Figures and Tables

**Figure 1 cancers-13-01658-f001:**
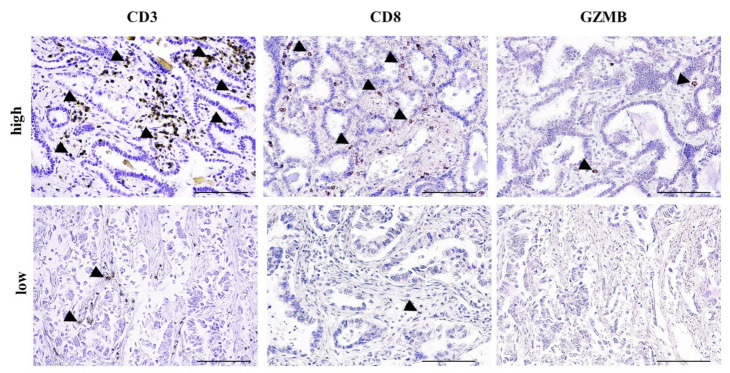
High-tumor-infiltrating lymphocyte (TIL) and low-TIL infiltrate. Representative photos with high and low CD3, CD8 and granzyme B (GZMB) infiltrate in two non-small cell lung cancer (NSCLC) patients. Arrowheads depict positive immunostaining. Scale bar: 100 μm.

**Figure 2 cancers-13-01658-f002:**
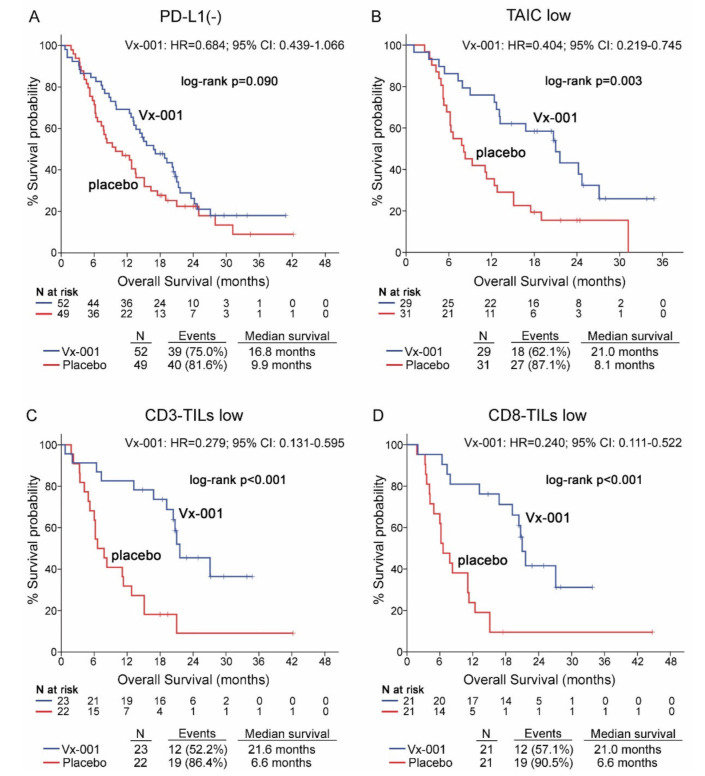
Overall survival (OS) in patients with PD-L1(-), tumor-associated immune cell (TAIC) low, CD3-TIL low and CD8-TIL low tumors. OS was measured in patients with PD-L1(-) (**A**), TAIC low (**B**), CD3-TIL low (**C**) and CD8-TIL low (**D**) tumors.

**Figure 3 cancers-13-01658-f003:**
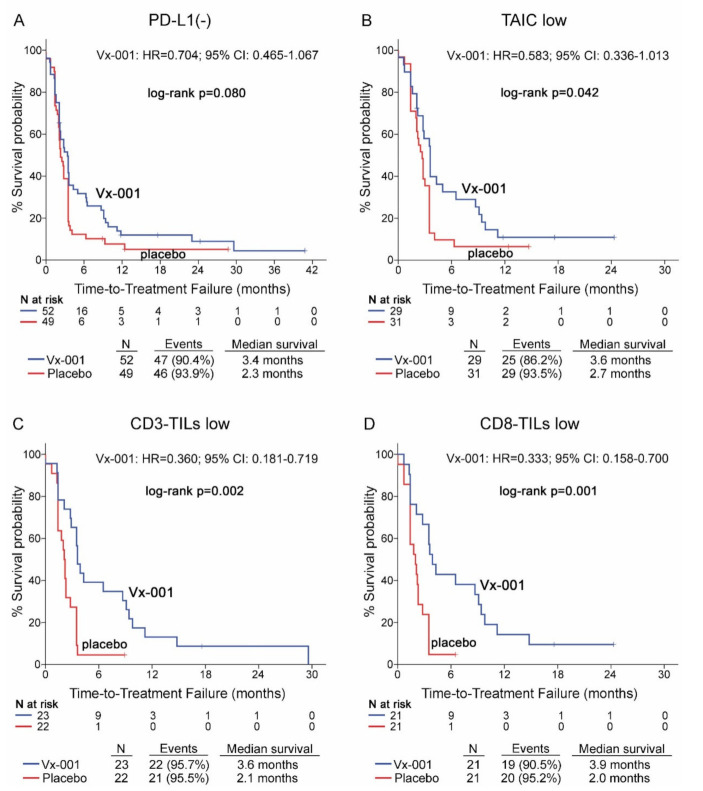
Time to treatment failure (TTF) in patients with PD-L1(-), TAIC low, CD3-TIL low and CD8-TIL low tumors. TTF was measured in patients with PD-L1(-) (**A**), TAIC low (**B**), CD3-TIL low (**C**) and CD8-TIL low (**D**) tumors.

**Figure 4 cancers-13-01658-f004:**
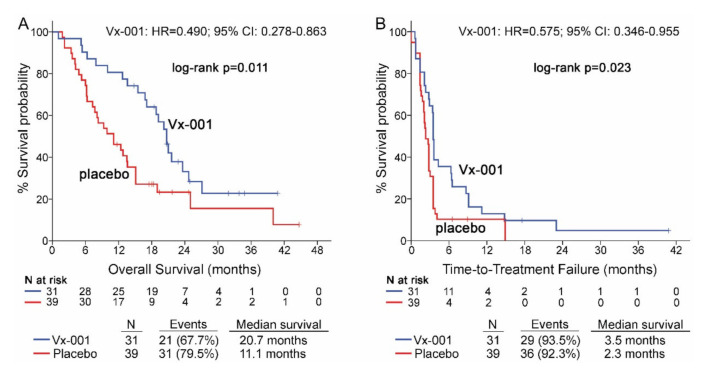
OS and TTF in patients with GZMB-TIL low tumors. OS (**A**) and TTF (**B**) were measured in patients with GZMB-TIL low tumors.

**Table 1 cancers-13-01658-t001:** Patients’ characteristics.

Variable	Total Patients, *n* = 131 No. of Patients (%)	Vx-001 Arm, *n* = 63 No. of Patients (%)	Placebo Arm, *n* = 68 No. of Patients (%)
**Gender**			
Male	95 (72.5%)	44 (69.8%)	51 (75.0%)
Female	36 (27.5%)	19 (30.2%)	17 (25.0%)
**Age**			
<65 years	63 (48.1%)	31 (49.2%)	32 (47.1%)
≥65 years	68 (51.9%)	32 (50.8%)	36 (52.9%)
**Histology**			
NSQ/MH	75 (57.3%)	34 (54.0%)	41 (60.3%)
SQ	56 (42.7%)	29 (46.0%)	27 (39.7%)
**Response to previous treatment**			
OR	65 (49.6%)	27 (49.2%)	38 (55.9%)
SD	66 (50.4%)	36 (57.1%)	30 (44.1%)
**Smoking status**			
Never	13 (9.9%)	6 (9.5%)	7 (10.3%)
Smokers	118 (90.1%)	57 (90.5%)	61 (89.7%)
Heavy smokers (>25 years)	93 (71.0%)	45 (771.4%)	48 (70.6%)
Light smokers (<25 years)	25 (19.1%)	12 (19.1%)	13 (19.1%)

NSQ/MH: non-squamous/mixed histology, SQ: squamous, OR: objective response, SD: stable disease.

**Table 2 cancers-13-01658-t002:** Univariate and multivariate Cox regression analysis of OS in the Vx-001 arm.

Univariate Analysis				
Covariant Tested vs. Control (HR = 1)	HR ^a^	95% CI ^b^	*p*-Value ^c^	Bootstrap *p*-Value ^c^
CD3-TILs Low vs. High	0.395	0.194–0.805	0.011	0.019
CD8-TILs Low vs. High	0.428	0.212–0.862	0.018	0.013
GZMB-TILs Low vs. High	0.310	0.122–0.792	0.014	0.048
TAIC Low vs. High	0.590	0.323–1.069	0.082	0.089
Gender Female vs. Male	1.398	0.753–2.595	0.288	0.285
Age ≥65y vs. <65y	0.963	0.543–1.706	0.896	0.894
Response to previous treatment SD vs. OR	1.349	0.748–2.433	0.320	0.306
Histology SQ vs. NSQ/MH	1.575	0.884–2.806	0.123	0.118
Smoking Heavy vs. No/Light	1.714	0.848–3.465	0.133	0.133
**Multivariate Analysis ^d^**				
CD3-TILs Low vs. High	0.381	0.181–0.799	0.011	0.015
CD8-TILs Low vs. High	0.388	0.182–0.830	0.015	0.016
GZMB-TILs Low vs. High	0.173	0.058–0.520	0.002	0.003

^a^ Hazard ratio; ^b^ 95% confidence interval of the estimated HR; ^c^ calculated by test for trend. Bootstrap *p*-value is based on 1000 bootstrap samples; ^d^ multivariate analysis adjusted for CD3-TILs, CD8-TILs or GZMB-TILs with patients’ gender, age, response to previous treatment, tumor histology and smoking status.

## Data Availability

The data presented in this study are available on request from the corresponding author.

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
