# Peer review of "Clinical Activity of an hTERT-Specific Cancer Vaccine (Vx-001) in “Immune Desert” NSCLC"

_cancers, 2021, doi:10.3390/cancers13071658_

Round 1
Reviewer 1 Report
The authors study the correlation between tumor vaccine Vx-001 clinical activity and the tumor microenvironment (TME). They show that the Vx-001 offers a clinical benefit in patients with tumors lacking or weakly infiltrated with tumor infiltrating lymphocytes (TILs) but not in the context of tumors highly infiltrated with TILs. In addition, TIL negative/low tumor signature is not a predictive factor of Vx-001 efficacy. Thus, they conclude that the selection of patients with TIL negative or low infiltrated tumors with bad prognosis to be the best candidates to show a clinical benefit from Vx-001 vaccination.
This report is interesting in clinical study. The inverse correlation between tumor vaccine efficacy and the presence of TILs is unique. However, the writing should be improved and some questions are listed below. Thus, I suggest that this manuscript should be revised.
Comments
- Line 35, I suggest to use “separated” instead of “distinguished”.
- I can’t agree with the statement in first paragraph of “Introduction”. The strategy should be finding neoantigen but not breaking tolerance, which will cause autoimmunity.
- The writing should be rearranged in Introduction section, too many short paragraphs.
- The subtitles “1.” Materials and Methods; “1.” Results; “1”., “1.1”….; “1.” Discussion; “1.” Conclusions?
- The authors clam that this is the first study showing an inverse correlation between tumor vaccine efficacy and the presence of TILs. They should discuss why Vx-001 has this property.
- Following question 5, please discuss how to apply such kind of vaccines in clinic.
Author Response
We made an effort to improve English language as required
Comments
- Line 35, I suggest to use “separated” instead of “distinguished”.
Answer: We followed the suggestion of the reviewer and replaced distinguished by separated
2. I can’t agree with the statement in first paragraph of “Introduction”. The strategy should be finding neoantigen but not breaking tolerance, which will cause autoimmunity.
Answer: We modified this paragraph as follows “Until now, cancer vaccines have largely failed because they targeted tumor self-antigens and were not able to overcome immune tolerance. [1] Overcoming tolerance to tumor antigens without inducing autoimmunity is therefore one of the main challenges in tumor vaccination”. However, we believe that the main property of neoantigens that are mutated self-proteins is that they escape the self-tolerance network because of their mutation(s).
3. The writing should be rearranged in Introduction section, too many short paragraphs.
Answer: We re-arranged the introduction section by removing one paragraph (lines 100-105).
4. The subtitles “1.” Materials and Methods; “1.” Results; “1”., “1.1”….; “1.” Discussion; “1.” Conclusions?
Answer: We changed the numeration of the segments
5. The authors clam that this is the first study showing an inverse correlation between tumor vaccine efficacy and the presence of TILs. They should discuss why Vx-001 has this property.
Answer: We believe that our findings could be applied to all cancer vaccine and not only to Vx-001 providing that these vaccines are able to overcome tolerance to tumor antigens and induce a strong tumor immunity. This point is now addressed in the revised Discussion section as follows:
“It is tempting to speculate that our findings with Vx-001 could be extended to all cancer vaccines that are able to induce a strong anti-tumor immunity. Thus, cancer vaccines should be tested in patients selected for their TIL low, non-immunogenic NSCLC tumors that are resistant to ICI and not in patients with TIL high, immunogenic NSCLC known to be sensitive to ICI”.
.
6. Following question 5, please discuss how to apply such kind of vaccines in clinic.
Answer: This is discussed in the revised Discussion section as follows:
“It is tempting to speculate that our findings with Vx-001 could be extended to all cancer vaccines that are able to induce a strong anti-tumor immunity. Thus, cancer vaccines should be tested in patients selected for their TIL low, non-immunogenic NSCLC tumors that are resistant to ICI and not in patients with TIL high, immunogenic NSCLC known to be sensitive to ICI”.
Reviewer 2 Report
The study by Pateras I. et al, demonstrates the efficacy of a therapeutic cancer vaccine (Vx-001) targeting a cryptic TERT peptide in HLA-A2+ patients with stage IV or recurrent stage I-III NSCLC. This study demonstrates for the first time that vaccination with Vx-001 is associated with an improved overall survival and time-to-treatment failure as compared to placebo in patients whose tumors were infiltrated with low numbers of TILs but not in patients with TIL-high tumors. Intratumoral analyses revealed that the Vx-001 vaccine improved clinical outcomes in patients whose tumors were infiltrated by low numbers of CD3-TIL, CD8-TIL or GZMB-TIL. The results from this study suggest that tumors with high levels of TIL infiltration have a suppressive microenvironment which inactivates the vaccine-induced mono- or oligo-clonal T cells whereas tumors with TIL negative/low infiltrated (non-immunogenic/cold) lack immunosuppressive microenvironments. This study is a part of a recently published phase II trial from the same group (ref. 9 in this manuscript). In this recently published study, the authors published data from unselected vaccinated patients whereas in the present submitted study these vaccinated patients were stratified by the levels of their TILs (ie. high vs low). Although the number of patients among groups is rather low, still in multivariate Cox models CD3-TIL low, CD8-TIL low and GZMB-TIL low emerged as independent predictive factors associated with improved overall survival in patients vaccinated with Vx-001.
To my opinion, this study is very important in that it paves the way for the future use of additional therapeutic cancer vaccines in patients with low TIL densities who are usually excluded from active immunotherapies.
There are some minor points which need to be addressed by the authors in order to make their conclusions more clear for the reader.
- The authors make the hypothesis that the Vx-001 vaccine is not effective in the immunosuppressive microenvironment of TIL-high immunogenic tumors. They explain that in fact this immunosuppressive microenvironment has been established due to different inhibitory mechanisms including the PD-1/PD-L1 pathway, probably as a result of immune resistance against the endogenous tumor (neo)antigen-specific immune response mediated by the high density TILs. Is there any other evidence to suggest that besides PD-1/PD-L1 additional immunosuppressive circuits including suppressor MDSC or Tregs do exist in the TIL-high tumors (dampening any vaccine-induce immunity)?
- In lines 305-307 (Discussion section) the authors mention that more than 50% of their GZMB-TIL low tumors were CD8-TIL high and yet they observed clinical efficacy with their vaccine. The authors should comment on this observation.
- In lines 193, 194 (Results section) the authors state that double CD8/GZMB staining, revealed that 27 of 30 patients’ tumors tested were negative for CD8(+)/GZMB(+) TILs. The authors should discuss the cell subset – source for GZMB expression.
- Table S4 presents overall survival data with patients having PD-L1(+), TAIC high, CD3-TIL high, CD8-TIL high or GZMB-TIL high tumors. How do the authors explain the reduced overall survival in the vaccinated group of patients vs the placebo group? (which in the GZMB-TIL high tumors reaches statistical significance?)
Author Response
- The authors make the hypothesis that the Vx-001 vaccine is not effective in the immunosuppressive microenvironment of TIL-high immunogenic tumors. They explain that in fact this immunosuppressive microenvironment has been established due to different inhibitory mechanisms including the PD-1/PD-L1 pathway, probably as a result of immune resistance against the endogenous tumor (neo)antigen-specific immune response mediated by the high density TILs. Is there any other evidence to suggest that besides PD-1/PD-L1 additional immunosuppressive circuits including suppressor MDSC or Tregs do exist in the TIL-high tumors (dampening any vaccine-induce immunity)?
Answer: We agree with the reviewer that MDSC and Tregs are important players in the induction of the immunosuppressive microenvironment. Unfortunately, we do not have data on the presence of MDSC and Tregs and the limited availability of slides from the tumor biopsies will not permit the proposed assessment
2. In lines 305-307 (Discussion section) the authors mention that more than 50% of their GZMB-TIL low tumors were CD8-TIL high and yet they observed clinical efficacy with their vaccine. The authors should comment on this observation.
Answer : We comment this point in the revised Discussion section as follows:
“It would be interesting to study additional markers of T cell activation such as IFNg and perforin to investigate if the activity of Vx-001 is related not only to the number of TILs but also to their functionality”
3. In lines 193, 194 (Results section) the authors state that double CD8/GZMB staining, revealed that 27 of 30 patients’ tumors tested were negative for CD8(+)/GZMB(+) TILs. The authors should discuss the cell subset – source for GZMB expression.
Answer: This point is discussed in the revised Discussion section as follows:
“The phenotype of GZMB(+)/CD8(-) TILs was not studied but it is very likely that they are NK cells”.
4. Table S4 presents overall survival data with patients having PD-L1(+), TAIC high, CD3-TIL high, CD8-TIL high or GZMB-TIL high tumors. How do the authors explain the reduced overall survival in the vaccinated group of patients vs the placebo group? (which in the GZMB-TIL high tumors reaches statistical significance?)
Answer: OS is not significantly shorter in Vx-001 treated than in placebo treated patients in the groups of TAIC high, CD3-TIL high and CD8-TIL high and significantly shorter in the group of GZMB-TIL high (15 patients). In contrast, TTF is comparable in Vx-001 treated and placebo treated patients in all these groups suggesting that there is no a deleterious effect of Vx-001. It is very likely that OS is shorter because the % of patients who received a subsequent treatment (2nd, 3rd etc lines) is higher in placebo than in Vx-001 treated patients in TAIC high (78% vs 64%), CD3-TIL high (84% vs 70%), CD8-TIL high (86% vs 70%) and GZMB-TIL (100% vs 57%).
Reviewer 3 Report
The manuscript by Pateras et al describes the activity of an hTERT-specific cancer vaccine, Vx-001, in non-small cell lung cancer (NSCLC) with low or no T-cell infiltration.
The clinical study was a randomized phase II trial in metastatic NSCLC with one vaccinated arm (63 patients) and one placebo arm (68 patients). Biopsies were retrospectively analyzed for PD-L1 expression and the presence of TIL using CD3, CD8 and Granzyme B as markers. No correlation was shown between PD-L1 and clinical effect of the vaccine, whereas TIL low patients in the vaccine treated group showed improved overall survival (OS) compared to the placebo group.
The authors hypothesize that this means that patients with low or non-immunogenic tumours can respond to the vaccine and there is no added benefit of the vaccine in patients that have already pre-vaccine responses and immunogenic tumours where the immune system has lost the capacity to control the tumour. The authors demonstrate a clear positive correlation between the lack of pre-treatment TIL and OS. CD3 and CD8 TIL measurements also correlated well and the lower presence of GZMB is discussed and referenced. The manuscript is well written, however, the conclusions and hypotheses would have been further strengthened if the authors had shown an induction of vaccine-specific responses in the vaccinated group versus the control arm.
The mechanism of the vaccine should be to induce peptide-specific T-cell responses, yet here no evidence of this is presented in the current manuscript.
Major points:
- The authors say that patients without a pre-vaccine response can have benefit of the vaccine, yet no vaccine-specific responses have been shown, only survival and pre-vaccination T-cell infiltration and PD-L1 expression. Which proportion of the patients had pre-vaccine responses and which method was used to measure it? The study describes that one of the outcome measures is frequency of TERT specific IFN-γ and perforin- producing T cells in the blood. How many of the patients with low or no TILs prior to vaccination displayed detectable post-vaccination immune responses in e.g. blood? This seems to correspond with the report in reference 9, Gridelli et al. 2020 where a survival benefit was shown for patients with an immune response to the vaccine. Hence the data should be integrated into the current manuscript where a subgroup of patients have shown to benefit and discussed in relation to the current findings.
- The authors say that one of the groups that would benefit from the vaccine is the non-smokers, but there are only 13 non-smokers included in the study. Would this be sufficient to conclude for this group? This seemed to be looked at for the same vaccine in reference 10, Gridelli et al 2017, but what were the clinical responses and the vaccine responses for the non-smokers in the current study?
- How about HLA expression in the tumours of patients with no TILs? The patients were selected for the HLA-A*02:01 allele to be eligible for vaccination with the peptide vaccine. Did the TIL low or negative tumours analyzed here still express HLA class I?
Minor:
Table 1, lines with response to previous treatment should be moved up to correspond to OR and SD
Author Response
Major points:
- The authors say that patients without a pre-vaccine response can have benefit of the vaccine, yet no vaccine-specific responses have been shown, only survival and pre-vaccination T-cell infiltration and PD-L1 expression. Which proportion of the patients had pre-vaccine responses and which method was used to measure it? The study describes that one of the outcome measures is frequency of TERT specific IFN-γ and perforin- producing T cells in the blood. How many of the patients with low or no TILs prior to vaccination displayed detectable post-vaccination immune responses in e.g. blood? This seems to correspond with the report in reference 9, Gridelli et al. 2020 where a survival benefit was shown for patients with an immune response to the vaccine. Hence the data should be integrated into the current manuscript where a subgroup of patients have shown to benefit and discussed in relation to the current findings.
Answer: As requested by the reviewer we
i). Added the immune response detection method in the revised Materials and Methods section
“Immune response
Vaccine induced immune response was evaluated before the first vaccination (baseline), before the third vaccination (W6) and at week 18 or at the end of treatment visit for patients dropped out before the sixth vaccination. Thereafter, from week 39, it was evaluated every 24 weeks.
IFNg ELISpot assay was performed using the Human IFN-g ELISpot PVDF-Enzymatic kit (Diaclone, Besançon, France) as described previously [9]”.
ii) Presented data on the correlation between immune response and TIL profile of the tumors in the revised Results section
“Vaccine induced immune response and TIL infiltration
TERT572 specific immune response was tested in 53 placebo treated and 52 Vx-001 treated patients. 17 (32.7%) patients in the Vx-001 arm developed immune response. Immune response was not detected in the placebo arm. There was no correlation between detected immune response and CD3/CD8-TIL profil of tumors. 40.9% (9/22) of patients with CD3/CD8-TIL low tumors and 25% (6/24) of patients with CD3/CD8-TIL high tumors responded to Vx-001 (p=0.348). Furthermore, the correlation of immune response and OS was investigated in patients with CD3/CD8-TIL low and CD3/CD8-TIL high tumors. Although the number of patients is low, the results suggested that immune response was correlated with longer OS in patients with CD3/CD8-TIL low but not in patients with CD3/CD8-TIL high tumors (Figure S4).
2. The authors say that one of the groups that would benefit from the vaccine is the non-smokers, but there are only 13 non-smokers included in the study. Would this be sufficient to conclude for this group? This seemed to be looked at for the same vaccine in reference 10, Gridelli et al 2017, but what were the clinical responses and the vaccine responses for the non-smokers in the current study?
Answer: When we analyzed the non-selected patient population (Gridelli et al., 2017) we found that Vx-001 was clinically active in never and light smokers (<25 years smoking history) (60 patients) [11.2 vs 20.2 months; HR=0.55; 95% CI 0.32-0.96; p=0.037] but not in heavy smokers (>25 years smoking history) (128 patients) [11.3 vs 13.1 months; HR=1.31; 95% CI 0.87-1.97; p=0.180]. In the current study we confirmed these results and showed that never and light smokers (38 patients) responded to Vx-001 [12.4 vs 24.2 months; HR=0.39; 95% CI 0.18-0.84; p=0.017] while heavy smokers (93 patients) did not [11.1 vs 14.7 months; HR=1.12; 95% CI 0.68-1.79; p=0.661]. However, we did not find any correlation between smoking status and infiltration with TAIC, CD3-TIL or CD8-TIL. We do not present these data in the current paper that is focused on the correlation between TIL infiltration and efficacy of Vx-001. We also modified Table 1 because there was an error on the number of light and heavy smokers in the current study.
3. How about HLA expression in the tumours of patients with no TILs? The patients were selected for the HLA-A*02:01 allele to be eligible for vaccination with the peptide vaccine. Did the TIL low or negative tumours analyzed here still express HLA class I?
Answer: Unfortunately, we do not have data on the HLA-I expression by tumors and the number of available biopsies to do this analysis is very low to draw a conclusion on the HLA-I expression in TIL low tumors in our patient population.
Minor:
Table 1, lines with response to previous treatment should be moved up to correspond to OR and SD
Answer: Lines are moved up to correspond to OR and SD
Round 2
Reviewer 3 Report
The authors have provided satisfactory answers to the questions and comments by the reviewer.
The new supplementary figure S4 shows that a low number of vaccine immune responders that had low TIL infiltration pre-treatment had improved OS compared to the immune responders with pre-treatment high TIL tumours. The text is in results section 3.5.
It would be very important to check specificity/additional markers of TILs in a follow-up study as well as post-treatment biopsies to see if an induction of an immune response increased post-treatment TILs. This could explain how the vaccination could lead to enhanced OS in the pre-treatment TIL low group as there is no mechanism currently provided for how the vaccine can induce a survival advantage in non-immune responders.
This manuscript is a resubmission of an earlier submission. The following is a list of the peer review reports and author responses from that submission.